# Hydrogen Trapping in bcc Iron

**DOI:** 10.3390/ma13102288

**Published:** 2020-05-15

**Authors:** Anastasiia S. Kholtobina, Reinhard Pippan, Lorenz Romaner, Daniel Scheiber, Werner Ecker, Vsevolod I. Razumovskiy

**Affiliations:** 1Materials Center Leoben Forschung GmbH, Roseggerstraße 12, 8700 Leoben, Austria; lorenz.romaner@mcl.at (L.R.); daniel.scheiber@mcl.at (D.S.); werner.ecker@mcl.at (W.E.); razvsevol@yahoo.com (V.I.R.); 2Department Materials Physics, University of Leoben, Jahnstraße 12, 8700 Leoben, Austria; 3Erich Schmid Institut of Materials Science, Austrian Academy of Sciences, Jahnstraße 12, 8700 Leoben, Austria; reinhard.pippan@oeaw.ac.at

**Keywords:** first principles calculations, hydrogen embrittlement, bcc iron, trapping energies

## Abstract

Fundamental understanding of H localization in steel is an important step towards theoretical descriptions of hydrogen embrittlement mechanisms at the atomic level. In this paper, we investigate the interaction between atomic H and defects in ferromagnetic body-centered cubic (bcc) iron using density functional theory (DFT) calculations. Hydrogen trapping profiles in the bulk lattice, at vacancies, dislocations and grain boundaries (GBs) are calculated and used to evaluate the concentrations of H at these defects as a function of temperature. The results on H-trapping at GBs enable further investigating H-enhanced decohesion at GBs in Fe. A hierarchy map of trapping energies associated with the most common crystal lattice defects is presented and the most attractive H-trapping sites are identified.

## 1. Introduction

Hydrogen embrittlement (HE) is a latent problem for structural materials, particularly for high-strength steels [1,2,3]. In spite of multiple existing theories of HE mechanisms [3], the actual cause of embrittlement remains unclear and requires further investigation in many cases. Density functional theory (DFT) offers a predictive methodology to study this problem at the atomic level and can provide answers to some important questions, such as H localization at defects and its impact on the interatomic bonding and cohesion in the system [4,5,6]. Furthermore, these data can be used to support or discard existing theories of HE; estimate the actual content of atomic H in the material in general and at the defects in particular; and assist in the interpretation of thermal desorption spectroscopy [7,8] data.

A significant amount of effort has been made with respect to investigations of H-trapping in Fe [9,10,11,12,13,14,15,16,17,18,19,20,21]. Most research works have focused on the identification of the most favorable position of the atomic H in the lattice or at a defect with the lowest trapping energy at 0 K. The data available in the literature suggest that the strongest traps for H at 0 K should be some special grain boundaries (GBs) [13,22]. Hydrogen trapping at special coincident site lattice (CSL) GBs has been described so far for the tilt Σ3(111), Σ5(012), Σ5(013) [13], Σ9(1/2 11), Σ13(1/3 11), Σ17(1/4 11) [15], and twist Σ3(110), Σ9(110), Σ11(110), Σ17(110) [14] GBs. Trapping energies as strong as −0.81 eV, −0.83 eV and −0.95 eV have been found for the Σ5(012)[100] tilt GB [13], and the Σ11(110) and Σ17(110) twist GBs [14], respectively. DFT studies on H-trapping at the edge and screw dislocations [16,17,23] suggest a range of the trapping energies from −0.19 to −0.47 [16,17,23]; i.e., sizeably lower values than those of the special GBs. Other DFT papers with a focus on H-trapping at vacancies and on formation of vacancy-H clusters in body-centered cubic (bcc) Fe report trapping energies varying from −0.4 to −0.6 eV [12,18,24], which are also lower than those of GBs. However, it is rather difficult to draw a definite conclusion on the hierarchy of trapping energies, as most of the aforementioned results have been obtained using different methods and approaches, which makes a quantitative comparison difficult or even impossible in some cases.

The problem gets even more involved when one starts considering not just the strongest trapping site, but the whole distribution of the trapping energies associated with a particular defect [25,26]. So far, only a few works have considered such distributions [4,27] and none have provided a systematic investigation of H-trapping energy distributions for the key defects using the same DFT methodology. Such an investigation is still required to evaluate the possible concentrations of H atoms at different defects at ambient temperatures relevant to the conditions at which HE occurs. 

The presence of traps in the bcc Fe significantly influences the diffusivity of H, and therefore is an important aspect of the problem of HE in Fe. This point has been thoroughly investigated in a number of DFT and molecular dynamics (MD) studies for each of the considered defects in our work. In the case of bcc Fe, the H diffusivity in the presence of traps has been reported to be significantly lower than in the Fe lattice. Lu et al. [28] reported that diffusivities were reduced as the point defect concentration increased, and the influence of such point defects as Fe vacancies and self-interstitial atoms reduces as the temperature increases. Lv et al. also found changes in the mechanism of H diffusivity at the presence of vacancies [29]. Kimizuka et al. showed that the H atom was strongly trapped at screw dislocations, and there is a high barrier for H diffusion both across and along the dislocation [30]. According to Teus et al., Fe GBs retard H migration [31]. Jiang et al. showed high diffusion barriers of H migration from surfaces Fe(110) (1.02 eV) and Fe(100) (0.38 eV) to subsurface layers and very small barrier for the reverse process (0.03 eV) [32]. Activation migration energies for H at all aforementioned defects were given in the literature: 0.037 [31], 0.024 [28], 0.088 [32] and 0.127 eV [33] for bulk; 0.232 eV in the presence of Fe vacancies at a concentration of 0.009% [28]; 0.314 eV at Σ5(013) [31]; and 0.43 eV along [111] direction in the system with screw dislocations [30]. The thermodynamics-based trap-diffusion model by Svoboda and Fischer [34,35] directly links the trapping energies and the trap densities to the diffusivity of H. They clearly presented how the diffusivity gets concentration dependent in the presence of traps. Drexler et al. recently applied this generalized Oriani approach for the model-based evaluation of thermal desorption experiments and related the results to DFT-calculated trapping energies [36,37].

In this work, we perform a systematic DFT investigation of H-trapping in the bulk lattice, at vacancies, dislocations and special GBs in ferromagnetic bcc Fe. In comparison to previous theoretical studies using different variations of the generalized gradient approximation (GGA) for the exchange-correlation functional or even less precise tight-binding (TB) approximation calculations, this study provides a consistent set of H-trapping energies obtained within the same methodological approach that provides a set of energies for qualitative and quantitative interpretation of experimental (for instance, thermal desorption spectroscopy) data. We propose to use the results of this investigation for a hierarchical analysis of H-trapping in iron and the qualitative comparison of the trapping energies at selected defects. In addition, we study trapping energy profiles near each of the defects and use these data to evaluate H concentration at the defects as a function of temperature within the framework of a classical segregation isotherm. Finally, we also provide insights into the impact of H-trapping on cohesion in Fe.

## 2. Computational Details

### 2.1. Electronic Structure and Total Energy Calculations

Spin-polarized calculations were performed within the framework of density functional theory (DFT) using the projector augmented wave method [38,39,40,41] as implemented in the Vienna Ab-initio Simulation Package (VASP) (5.4.1, Materials Center Leoben Forschung GmbH, Leoben, Austria) [40,42]. Exchange-correlation effects were treated using the generalized gradient approximation (GGA) employing the Perdew, Burke and Enzerhof (PBE) [43] scheme. The convergence criteria were, for the total energy, 10^−5^ eV, and for the forces, 10^−3^ eV/Å. Ionic relaxations were included in all calculations. The cell shape and volume were kept fixed during the relaxations using the 0 K equilibrium volume of Fe unless specified otherwise. The calculations were performed using a plane-wave cutoff energy of 400 eV. The integration over the Brillouin zone was performed using the Monkhorst–Pack [44] meshes described in the next section in more detail. The VESTA software package [45] was used for visualisation of the atomic structures.

### 2.2. Structure Models

#### 2.2.1. Bulk

The equation of state was fitted by the Birch–Murnaghan equation [46,47] for calculation of the Fe bcc lattice parameter and bulk modulus. The supercells of 16, 54 and 128 atoms were employed to investigate H solution energies in the bulk of bcc iron. The structure models were prepared as the 2 × 2 × 2, 3 × 3 × 3, 4 × 4 × 4 replications of the two-atom conventional bcc cell. In all VASP calculations, 6 × 6 × 6, 4 × 4 × 4 and 4 × 4 × 4 k-point meshes were used for each of the aforementioned supercells respectively. Hydrogen has three possible sites in the bcc Fe lattice: (i) an octahedral interstitial; (ii) a tetrahedral interstitial and (iii) a substitutional site (see Figure 1).

#### 2.2.2. Interfaces

Special CSL model GBs Σ3(111) [1,2,3,4,5,6,7,8,9,10], Σ5 (012) [100] and Σ5 (100) [001] were modelled by supercells containing 49, 30 and 44 atomic layers of Fe (two, one and five atoms per layer) separated by 15, 7 and 7 Å of vacuum, which were tested to be sufficient within 0.01 eV/at error at most, as schematically shown in Figure 2.

The same supercells, but without GB, have been used for the (111), (012) and (100) free surface (FS) calculations. The 6 × 4 × 1, 14 × 6 × 1, 2 × 2 × 1 Monkhorst-Pack k-point meshes were used for GB and FS calculations. The structures of the Σ3(111) [1,2,3,4,5,6,7,8,9,10,26], Σ5 (012) [100] [48] and Σ5 (100)[001] [26]GBs were relaxed by shifting two grains in the slab with respect to each other. The discovered minimum energy structures were used in all GB slab calculations. Hydrogen atoms were inserted in the interstitial positions one at a time in the first three GB/FS layers starting from the GB/FS plane, as shown in Figure 2a–c (I0–I2). In the case of (120) and (100) FS, only the FS layer (i0) has been considered for H segregation.

#### 2.2.3. Dislocations

Two dislocations were considered in this work: (i) the ½<111> screw dislocation and (ii) the mixed M111 dislocation, wherein the Burgers vector and dislocation line are along non-parallel [111] directions intersecting at an angle of about 70.5°. Dislocations were treated in 3D periodic structures. For the case of the screw dislocation, the quadrupole arrangement was used which had already been proven to reliably describe core structures, energies, and Peierls stresses [49,50,51,52]. For the M111 dislocation, a suitable periodic geometry was considered. In both cases, two dislocations with antiparallel Burgers vectors were inserted into unit cells characterized by the following lattice vectors: $a_{1}=5u_{\left[11\bar{2}\right]},a_{2}=9u_{\left[\bar{1}10\right]}+½u_{\left[111\right]}$ and $a_{3}=u_{\left[111\right]}$ in the screw dislocation case, and $a_{1}=4u_{\left[11\bar{2}\right]},a_{2}=11u_{\left[\bar{1}10\right]}$ and $a_{3}=u_{\left[111\right]}$ in the mixed dislocation case, where $u_{\left[11\bar{2}\right]},u_{\left[\bar{1}10\right]},u_{\left[111\right]}$ were basis vectors, connecting two atoms of the bcc lattice along the specified direction [53]. The resulting supercell geometries included 135 and 253 atoms, as shown in Figure 3. The k-point meshes were 1 × 2 × 16 and 1 × 1 × 16 for the screw and mixed dislocation respectively, which proved to yield convergent results in earlier works [54,55].

## 3. Methodology

### 3.1. Solution Energies

The solution energy of the substitutional and interstitial H is defined as:(1)$$\Delta E_{sol}^{sub}=E_{sc}[N-1;1]-\frac{N-1}{N}E_{sc}[N;0]-\frac{1}{2}E_{H_{2}}$$
(2)$$\Delta E_{sol}^{\mathrm{int}}=-\Delta_{bulk}^{H}-\frac{1}{2}E_{H_{2}}$$
(3)$$\Delta_{bulk}^{H}=E_{sc}[N;0]-E_{sc}[N;1]$$
where $E_{sc}[n;m]$ represents the total energy of a bulk supercell, containing *n* host atoms and m H atoms; $E_{H_{2}}$ is the total energy of the H molecule in its equilibrium (fully relaxed) configuration at 0 K; $\Delta_{bulk}^{H}$ is the energy difference between the pure bulk supercell and bulk supercell after H atom is added.

### 3.2. Hydrogen Trapping at Defects

The energy of H-trapping by a vacancy, a dislocation, the FS and a GB at interstitial positions is defined as:(4)$$E_{trap}^{def}=E_{sc}^{def}[m]-E_{sc}^{def}[m-1]+\Delta_{bulk}^{H}$$
where $E_{sc}^{def}[m]$ and $E_{sc}^{def}[m-1]$ represent the total energies of supercells, containing one of the defects (vacancy, dislocation, GB, FS) and m and m-1 H atoms respectively.

We note that the trapping energies of Equation (4) are defined so that a negative energy means the energetically favored trapping. When we compare several negative trapping energies using “lower” and “higher” wording, then we mean a more negative, i.e., a more trapped energy in the first case, and a less negative, i.e., a less trapped energy in the second case.

### 3.3. Effect of H on the Bulk Cohesive Strength

The partial cohesive energy *χ_i_* is a fundamental quantity that can be used to characterize the effect of H (with concentration *c_i_*) on the cohesive strength of the bulk of an alloy [56]. In this work, the partial cohesive energy *χ_i_* is calculated as: (5)$$\chi_{i}=\frac{\partial E_{coh}}{\partial c_{i}}{}_{c=o}=\Delta_{bulk}^{H}+(E_{coh}^{H}-E_{coh}^{0})+E_{cryst}^{H}$$
(6)$$E_{coh}=E_{at}-E_{cryst}$$
where $c_{i}$ is the impurity concentration. $E_{coh}^{i}$ and $E_{coh}^{0}$ are the cohesive energies of the impurity and host species, $E_{cryst}^{i}$ is the calculated total energies (per atom) of impurities in their respective most stable crystalline phases; $E_{at}$ is energies of an isolated atom. All these energies are calculated by DFT at 0 K. The bulk supercell chosen for this calculation contains 128 atoms.

### 3.4. Effect of Trapping on GB Cohesive Strength

The ideal work of separation, $W_{sep}$ is a fundamental thermodynamic quantity that controls the mechanical strength of an interface [57] and can be defined as:(7)$$W_{sep}^{0}=2\gamma_{fs}^{0}-\gamma_{gb}^{0}$$
where $\gamma_{fs}^{0}$ is the surface formation energy and $\gamma_{gb}^{0}$ is the GB formation energy, which can be obtained by:(8)$$\gamma_{fs}^{0}=(E_{slab}^{fs}[N;0]-N_{slab}^{fs}\varepsilon_{slab}^{0})/2A$$
where $E_{slab}^{fs}$ is the total energy of a slab containing the FS, $N_{slab}^{fs}$ is the number of atoms in the supercell, $\varepsilon_{slab}^{0}$ is the total energy of the space filling slab (the slab supercell of the same geometry as used for $E_{slab}^{fs}$ calculations completely filled up with layers of Fe) divided by the number of atoms and A is the cross sectional area of the supercell. The factor of two arises from the fact that there are two FSs per supercell [58]. The supercell, containing a GB, also includes two FSs. If FSs and GBs are chosen so that they contain equal numbers of atoms, we can define the GB energy as:(9)$$\gamma_{gb}^{0}=(E_{slab}^{gb}-E_{slab}^{fs})/A$$
where $E_{slab}^{gb}$ is the total energy of the supercell containing the GB and 2 FS (see Figure 2).

The strengthening energy *η_i_* is a quantity that characterizes the change of the work of separation into two FSs by changing the impurity excess from 0 to $\Gamma_{GB}$. Within the framework of the Rice–Wang theory [59], in the fast separation limit, it can be used to evaluate the propensity of a material toward brittle fracturing. Here, we adopt the following sign convention: a positive value of *η_i_* corresponds to GB embrittlement (reduction of work of separation) and a negative value to cohesive strengthening (increase in the work of separation). In the case wherein the same slab geometries are used in the calculations of the FS and GB, strengthening energy *η_i_* can be expressed as follows [58]:(10)$$\eta_{i}=\frac{W_{sep}^{0}-W_{sep}^{i}}{\Gamma_{GB}}=E_{trap}^{gb}-E_{trap}^{fs}$$
where $W_{sep}^{i}$, $W_{sep}^{0}$ are the work of separation energies with and without H; $\Gamma_{GB}$ is the H excess; $E_{seg}^{gb}$ and $E_{seg}^{fs}$ are the H segregation energies of the GB and FS, respectively.

### 3.5. Determination of H Concentration from Segregation Energies

The interface trapping energies at 0 K described in the previous section can be directly used to estimate impurity concentration at the GB at T > 0 K within the framework of the McLean–Langmuir isotherm assuming an ideal mixture of the solute and solvent atoms [60]:(11)$$\frac{c_{k}}{1-c_{k}}=\frac{c_{0}}{1-c_{0}}\exp\left(-\frac{E_{trap}^{gb}(k)}{k_{B}T}\right)$$
where $c_{k}$ is the occupancy of a GB site k by solute atoms, $c_{0}$ is the site occupancy by solute atoms in the bulk, $k_{B}$ is the Boltzmann constant and T is temperature.

The effective GB trapping energy can be written down as: (12)$$E_{trap}^{gb}(k)=k_{B}T\left[\ln\left(\frac{c_{0}}{1-c_{0}}\right)-\ln\left(\frac{\langle c_{k}\rangle }{1-\langle c_{k}\rangle }\right)\right]$$
where $\langle c_{k}\rangle =1/N\sum_{k}c_{k}$ is averaged over *k* trap sites within the trap width δ [25]; i.e., the spatial interval with no zero $E_{trap}^{gb}(k)$. Here we assume δ to be equal to the first 3 nearest neighbor distances from the defect. The corresponding effective solute concentration at the GB can be then expressed as:(13)$$<c_{k}>=\frac{c_{0}\exp\left(-\frac{E_{trap}^{gb}(k)}{k_{B}T}\right)}{1-c_{0}+c_{0}\exp\left(-\frac{E_{trap}^{gb}(k)}{k_{B}T}\right)}$$

The multi-site McLean–Langmuir isotherm used in this study accounts for configurational temperature effects only, while phonon and magnon contributions to the free energy of segregation are neglected. Since the temperatures of practical interest for the HE problem lay around and below the room temperature, we believe that this approximation represents a reasonable choice within this limit, as detailed phonon and magnon investigations for Fe-H system represent a formidable task at the moment and go beyond the scope of this paper.

## 4. Results

### 4.1. Pure Fe

#### 4.1.1. Bulk

##### Ground State Properties of the Bulk Fe

The 0 K equilibrium lattice constant and the bulk modulus of ferromagnetic (FM) bcc Fe along with the spin magnetic moment are presented in Table 1. The calculated lattice constant of 2.831 Å is in good agreement with other DFT works [33,61,62,63]. This value underestimates the experimental value of 2.853 Å extrapolated to 0 K [63]. This is a general result of DFT calculations of Fe and its alloys [64,65,66,67,68]. The calculated bulk modulus of 181 GPa agrees well with other DFT paper using the same exchange-correlation functional as used in this work [33,61,62] and slightly overestimates the low-temperature experimental value of 173 GPa [69]. The calculated magnetic moment on Fe of 2.19 µ_B_ has also been found to agree well with reported in literature values of 2.20–2.33 µ_B_ [33,61,62,70].

##### Vacancy

Vacancies belong to one of the most common thermal defect types considered in the literature in relation to the problem of HE. They play an important role in diffusion processes in Fe [33,72,73] and in trapping of H atoms [1,13]. The formation energy of a vacancy in FM Fe at 0 K is shown in Table 2. Our value of 2.02 eV falls in the midrange of DFT results available in literature [18,72,74,75,76,77,78,79] and agrees well with the experimental data on the vacancy formation in the FM Fe [80]. The scatter in the DFT values is related to the pronounced dependence of the vacancy formation energy upon changes in the lattice parameter, the magnetic state (ferromagnetic/paramagnetic) and the exchange-correlation energy potential.

##### Dislocation

Dislocations are another important type of defect in iron, as they play a crucial role in plastic deformation processes in the material. Therefore, H-trapping at dislocations represents an important task for understanding the effect of H on the plasticity in iron and its alloys [81,82,83,84,85]. DFT modelling of dislocations is a very challenging and formidable task in many cases due to the physical dimensions of the defect (10³–10^4^ atoms can be required to model such a defect). However, there are some methods that allow one to model dislocations of some special types [49,50,51,52,86]. In this work, we focus on two types of dislocations; namely, (i) ½<111> screw dislocation created using the periodic quadrupole arrangement [49], and for (ii) M111 mixed dislocation, a suitable periodic geometry was considered. The atomic structures of these dislocations after relaxation are shown in Figure 3. The [111] component of the relative displacement of the neighboring atoms produced by the dislocation is depicted as an arrow between them. The lengths of the arrows are proportional to the relative shifts of two neighbouring atoms along the surface normal, when inserting the dislocation in the perfect crystal. An arrow, connecting two neighbouring atoms, represents a shift of 1/3 Burgers vectors [87]. The dislocation geometry is illustrated by a differential displacement map [88] in Figure 4a,b. The screw dislocation exhibits a compact core, as expected [49,55,89,90]. The M111 dislocations exhibits a planar core, as discussed in the seminal work by Vitek [88].

Note that in contrast to the other defects, there is no characteristic defect energy that can be provided for the case of dislocations. This resides in the fact that the line energy diverges logarithmically, and therefore, the formation energy shows no convergence with the system size. The core energy, which would converge and could be obtained by subtracting the linear elastic energy, depends on the arbitrary choice of core radius and elastic constants, and hence, is also not characteristic for the dislocations.

#### 4.1.2. Interface

##### Grain Boundary and Free Surface

Intergranular cleavage failure has been experimentally observed in various materials exposed to H-rich environments [91,92,93]. Therefore, GBs are often seen as one of the central microstructural elements in investigations of HE. In this work we consider three types of special CSL GBs: tilt Σ3(111) [1,2,3,4,5,6,7,8,9,10], tilt Σ5 (012) [100] and twist Σ5 (100) [001] GBs which upon brittle fracture cleave into (111) (021) and (100) FS respectively. The associated work of separation, and the formation energies of the aforementioned interfaces are presented in Table 2. Comparison of the calculated results to the analogous literature data yields good agreement between the current DFT [64,80,94,95,96,97] results found in literature. The Σ3(111) [1,2,3,4,5,6,7,8,9,10], Σ5 (012) [100] and Σ5 (100) [001] GBs have GB energies of 1.60, 2.01 and 2.01 eV/at, respectively.

### 4.2. Iron + Hydrogen

#### 4.2.1. Hydrogen Trapping in the Bulk

##### Hydrogen Solubility in Fe Lattice

We have considered three possible sites for H dissolution in the Fe lattice: (i) the tetrahedral interstitial, (ii) the octahedral interstitial and (iii) the substitutional site. In agreement with literature data [62,78,79,107,108], our calculations show that the most favourable site for H in bcc Fe lattice is the interstitial tetrahedral with a formation energy of 0.23 eV. With the chosen plane-wave cutoff and k-point sampling, the reported formation energies are estimated to be converged. The interstitial octahedral and the substitutional sites have higher formation energies of 0.37 and 2.54 eV, respectively (see Table 3). These results have been obtained using the largest supercell (SC) of 128 atoms considered in this work. However, as it has been shown in reference [109], the formation energies of point defects may have a very slow convergence with respect to the supercell size, and therefore an extrapolation may be required to get an accurate value of the formation energy in the dilute limit. Following the methodology of reference [109], we have calculated the solution energy of H at the most stable interstitial tetrahedral position as a function of the supercell size N at the constant 0 K equilibrium volume of bcc Fe (allowing only for relaxation of the atomic positions) and at constant zero pressure (allowing for the complete relaxation of the atomic positions, SC shape and volume). In the limit of 1/N → 0 (infinitely large SC), these two quantities converge to a single value [11] corresponding to the “true” dilute limit.

Our results demonstrate this type of behaviour resulting in single value of the solution energy of H of 0.22 eV. The results are shown in Figure 4. A direct comparison of the present DFT 0 K results to the available experimental ones [107,110,111,112] requires a correction to the zero point vibrational energy (ZPE) that has been calculated to be equal to 0.10 eV (in the case of tetrahedral site TS of H) [62]. The ZPE-corrected experimental data (shown in Figure 5) were found to be in very good agreement with the present DFT results.

##### Hydrogen Trapping at A Vacancy

Hydrogen trapping at a vacancy has drawn a lot of attention in the literature [12,24,74,114]. It is known that there are six potential octahedral, interstitial trapping sites for H adjacent to the vacancy [24]. In references [18,24], it was found that H may form a stable complex with a vacancy in Fe consisting of two H and one vacancy. Here, we focus on the lowest energy cluster configurations found in reference [12] (1H-V, 2H-V and 3H-V). The results presented in Figure 6 and Table 4 confirm that the 2H-V cluster has the lowest trapping energy equal to −0.63 eV/at, stronger than the −0.61 value [12,24,114] found in other works. The results also show that there is a strong H-vacancy attractive interaction for nH-vacancy clusters with n < 6. The most stable 2H-vacancy cluster has a trapping energy of −58 eV/at, which is close to the 1H-V complex, whereas all other complexes have substantially higher trapping energies. Addition of the ZPE correction to the trapping energies based on available literature data reduces the trapping energies by about 0.11 eV [62] for 1H-V, 2H-V complexes and by 0.04 eV [62] for 3H-V (see Figure 6).

The experimental results on deuterium trapping reported in references [115,116] suggest that there are two sorts of −0.48 and −0.63 eV trapping energies associated with H-trapping at vacancies. In reference [115], the lowest energy trap −0.63 eV has been associated with the 1H-V, 2H-V defect complexes, whereas the other trapping energy of −0.48 eV has been related with the (3–6)H-V complexes based on the conclusions drawn from the effective-medium theory calculations [115]. The results of the effective-medium theory calculations [115] are available relative to 1H-V cluster only, and are therefore shown relative to the lowest experimental trapping energy of −0.63 eV in Figure 6, as in the original paper [115]. This interpretation of the experimental data agrees in general with our results and the theoretical literature data presented in Table 4. And Figure 7 shows hydrogen trapping profiles for Σ 3 (111) GB, (111) FS, mixed 111 dislocation and a vacancy.

##### Hydrogen Trapping at a Dislocation

Two types of dislocations, the ½<111> screw and the M111 mixed have been considered for H-trapping in Fe. Hydrogen segregates to dislocation core structure in the case of both considered dislocations. In addition to the displacement related to the dislocation core insertion to the bcc Fe lattice, shown in Figure 4, larger displacements appear near to the H location in cases of the screw and mixed dislocations (see Figure 8).

The hydrogen atom has been inserted in the cell with the screw dislocation, as illustrated in Figure 3a. The corresponding minimal segregation energy is listed in Table 4. In agreement with reference [17] the energetically most favourable position has been found to be located in the three corners of the screw dislocation core, as shown in Figure 3a. It should be noticed that the displacements are significantly amplified next to the H atom, which illustrates attractive interactions between H and the dislocation.

In the case of the M111 dislocation, H was placed at several positions in the core according to Figure 8b–f. The hydrogen was moved normal to the glide plane, as this does not induce dislocation glide. Attempts were made to place the H atoms on the glide plane too, at some distance from the dislocation. In this case, the dislocation followed the H atom, indicating that the positive attraction is strong enough to overcome the Peierls barrier. In what follows, these positions were not taken into account. The segregation energies are provided in Table 4. In Figure 8, the relaxed geometries with the H atom in positions 0 and 1 are shown. In these cases, H breaks the symmetry of the core structure. The displacements were amplified next to the H atom in agreement with the observation for the screw dislocation of an attractive interaction. The strongest segregation site was located close to the centre of the dislocation but not exactly in the middle. Similarly to the screw dislocation and the vacancy, the energetically most favourable position was not in the centre of the dislocation. In general, segregation energies are lower for the M111 dislocations compared to the screw dislocations. It should be taken into account that the segregation energy profile at the M111 dislocation core is not symmetric, i.e., site −2 is not equal to site 2 (see Figure 3b), since the lattice is compressed for negative segregation site indices and expanded for positive segregation site indices (Figure 9). This reveals that the solubility increases in expanded regions and decreases in compressed regions (Figure 3b), as one would intuitively expect.

The sites with the minimum trapping energies of −0.21 and −0.37 eV are found for H located at the corner of the dislocation core for the ½<111> screw and near to the centre of the dislocation core (marked with triangle in Figure 8) for the M111 mixed dislocations, respectively. As one can see in Figure 7 and Figure 8, H is repelled from the “compressed” region near the M111 mixed dislocation core and attracted to the “expanded” one. In the case of the screw dislocation, the trapping energies of the corner geometries (see Figure 3) have similar values of −0.21 eV and are in good agreement with the results of Itakura [17] (see Table 4).

#### 4.2.2. Interfaces

##### Hydrogen Trapping at GB

We have considered several possible sites for H-trapping in the GB planes of three special CSL GBs—CSL Σ3 (111) [1,2,3,4,5,6,7,8,9,10], Σ5 (012) [100] and Σ5 (100) [001] GBs—as shown in Figure 2. In the case of Σ3 GB, six equivalent positions of H (see Figure 2a) refer to octahedral site OS in the original bcc lattice and correspond to the strongest H-trapping energies ($E_{trap}^{gb}$ = −0.48 eV). This result has been found to be in good agreement with the literature [10,13,27,125]. The hydrogen atoms in the t position located in the first layer after the GB plane are not stable and relax to the OS positions ($E_{trap}^{gb}$ = −0.36 eV). The hydrogen atom placed in t position (see the Figure 2a), corresponding to TS in the original bcc lattice, is located in the next layer after Σ3 GB plane layers ($E_{trap}^{gb}$ = −0.12 eV).

In the case of Σ5 (012) [100], four inequivalent positions of H in the GB plane have been considered (see Figure 2b) using the Voronoi tessellation [125] for the identification of possible segregation sites. The lowest trapping energy belongs to the 4i site ($E_{trap}^{gb}$−0.42 eV), while others have the trapping energies from −0.34 to −0.37 eV.

For the twist Σ5 (100) [100] GB, nine inequivalent positions of H within ±6 Angstrom from the GB layer are found, using the Voronoi tessellation [125]. The strongest trapping energy of H has been found at site 1i shown in Figure 2c.

In the case of H-trapping at Σ3 (111) [1,2,3,4,5,6,7,8,9,10] GB, we have additionally considered the possibility of H atoms segregating off the GB plane. We have considered the same lowest energy trapping site for the near GB layers as 3t site (see the Figure 2a) in all calculations. The calculations (see Figure 7) have shown that the lowest energy site located beyond the GB plane has always been the 3t site, as shown in Figure 2a.

##### Effect of H on the Bulk and GB Cohesion

Hydrogen presence in the lattice can deteriorate the interatomic bonding in the crystal. Here, we use the partial cohesive energy (χ) concept [56] to evaluate the influence of H on the cohesive strength in the bulk of Fe. The results of our DFT calculations, shown in Table 5, provide us with a negative value of χ of −3.34 eV/at, indicating that H will deteriorate the interatomic bonding in the Fe lattice and reduce its resistance to decohesion.

We have investigated the effect of H-trapping on the GB cohesive strength in Fe using the strengthening energy η. For that purpose, H-trapping at FS in Fe in the positions shown in Figure 2 has been calculated as well (H sites at (111), (012), (001) FS correspond to 3t, 4i and from 1i positions from Figure 2a–c). The results of the H-trapping profile calculations at Σ3 (111) GB (Figure 7) have shown that the most favorable among considered for segregation sites is located within the interface plane. In the case of all (111), (012) and (001) FS created by cleavage of Σ3 (111) [1,2,3,4,5,6,7,8,9,10], Σ5 (012) [100] and Σ5 (100) [001] GBs respectively, we have used the in-plane FS sites to calculate the strengthening energy η. As one can see from the results shown in Table 5, H embrittles all GBs and has the η values varying from 0.05 (Σ5 (100) [001]) to 0.41 eV (Σ5 (012) [100]).

## 5. Discussion

### 5.1. Trap Hierarchy at 0 K

Comparison of the H-trapping energies at 0 K presented in Table 2 allows us to split the traps into two groups: (i) vacancies and GBs with the associated trapping energies varying from −0.39 to −0.63 eV and (ii) dislocations with the trapping energies < −0.37 eV. Our results show that H-trapping at the considered defects is described by a distribution of trapping energies shown in Figure 6 rather than by single trapping energy values used in most of the works on H-trapping so far [13,14,16,17,18]. The absolute lowest trapping energy of −0.63 eV has been found for the case of the vacancy-2H cluster. This trapping energy is followed by the H-trapping energy at the Σ5 (100) twist GB, which is only 0.06 eV higher. Following the results of reference [14], we are prone to think that twist GBs with higher Σ values (mostly not feasible for DFT investigations) could exhibit even lower trapping energies, which makes GB virtually equivalent to vacancy traps in Fe. Therefore, we have assigned GBs and vacancies to one group of defects with very similar trapping energies for H atoms, which could be indistinguishable in experiments. In general, H is found to be trapped at all considered types of defects and to have a negative effect on the cohesive strength of interatomic bonding in both bulk and interfaces.

### 5.2. Traps at Finite Temperatures

The effective H concentration at the vacancy, M111 dislocation and Σ3 (111) GB determined using Equation (13) is shown in Figure 9. Here, we have assumed 100 at ppm (a) in Figure 9 and 1 at ppm (b) in Figure 9. H content in the bulk of Fe [25,26,126,127] and the 0 K H-trapping profiles are shown in Figure 7.

The results of the McLean–Langmuir segregation isotherm at 100 at ppm H in the bulk of Fe suggest that H predominantly occupies GB and vacancies at low temperatures (<100 K). In the temperature interval from 100 to 400 K, H concentration at all considered defects is virtually the same with a slight preference to the M111 dislocation, whereas most Hs are accumulated at vacancies at T > 400 K followed by the GB and dislocations. At the room temperature +/-100 K (approximately 200–400 K), H concentration at dislocations shows a much faster decreasing tendency (at both 1 and 100 ppm H in the bulk), while H concentration at vacancies and GBs remains basically unchanged. It is related to the different shape of the H-trapping profile for the dislocation in comparison to H-trapping profiles at the GB and vacancy shown in Figure 7. This result indicates that H-trapping at the M111 type of dislocations can be more sensitive to the bulk H concentration and temperature changes than trapping at GBs and vacancies, which is an important aspect of the HELP mechanism of HE.

The overall amount of H in the system dramatically drops as temperature increases. Concentration of H at almost all defects decreases by a factor of 5 (GB and vacancy) or 3 (dislocation) as the temperature increases from 0 to 400 K. At 1 at ppm H in the bulk of Fe the H concentration at dislocation has significantly dropped but it has remained at about the same level as it was for GB and vacancy, assuming infinitely large grains and the amount of vacancies proportional to the concentration of H. These results show that H can be evenly distributed between different defects at the room temperature +/−100 K. This would also mean that the effective H-trapping energies at vacancies, GBs or dislocations can be virtually indistinguishable from one another and the interpretation of some experimental and theoretical results in terms of preferred trapping sites should be done with extra caution.

## 6. Conclusions

Hydrogen trapping in the bulk lattice and at all typical defects in bcc Fe has been systematically investigated by means of the same methodology of DFT calculations at 0 K. The results show that H occupies the tetrahedral interstitial site in the bulk lattice and prefers trapping at GB and vacancies to trapping at screw and mixed dislocations at 0 K. The mixed dislocation has been found to be a more attractive trap for H in Fe than the screw. Our results also show that trapping energies at each defect represent a distribution of trapping energies rather than a single trapping energy value.

We have used these unique sets of trapping energy distributions to evaluate H concentration at all considered defects at finite temperatures using the McLean–Langmuir segregation isotherm. The results of the segregation isotherm modeling using DFT trapping energy profiles suggest that all considered defects may have virtually the same amount of trapped H atoms at about room temperature +/−100 K, and therefore are equally important for addressing the problem of HE in Fe. This result also indicates that a special care has to be taken for interpretation of experimental data on H-trapping at room temperatures using DFT results obtained at 0 K. DFT calculations of the partial cohesive and the GB strengthening energies suggest that H will have a negative effect on the cohesive strength of interatomic bonding in both bulk and at the interfaces in bcc Fe.

## Figures and Tables

**Figure 1 materials-13-02288-f001:**
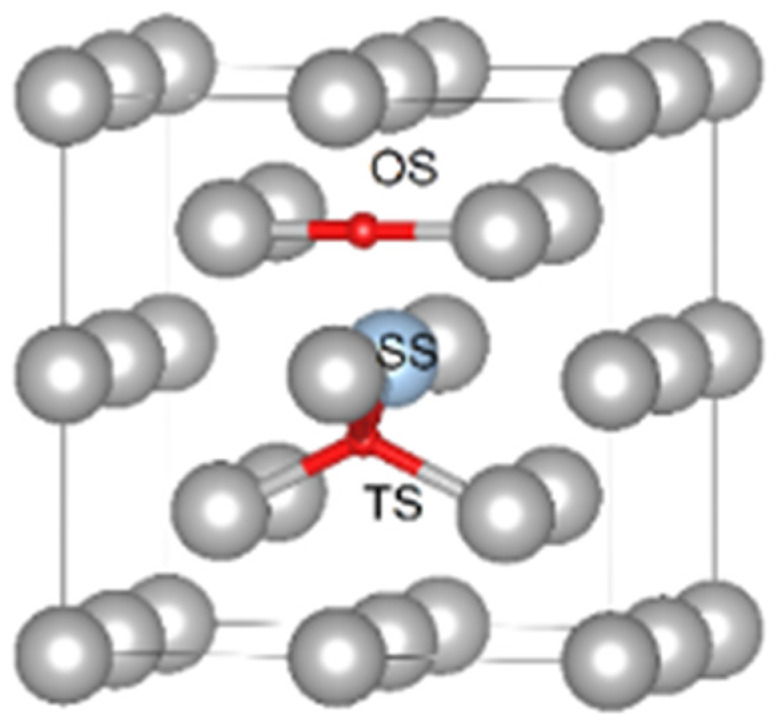
Possible sites for H in the bulk of bcc Fe. The red spheres correspond to the interstitial positions (OS, TS correspond to the octahedral and tetrahedral sites) and the blue sphere corresponds to the substitutional site (SS).

**Figure 2 materials-13-02288-f002:**
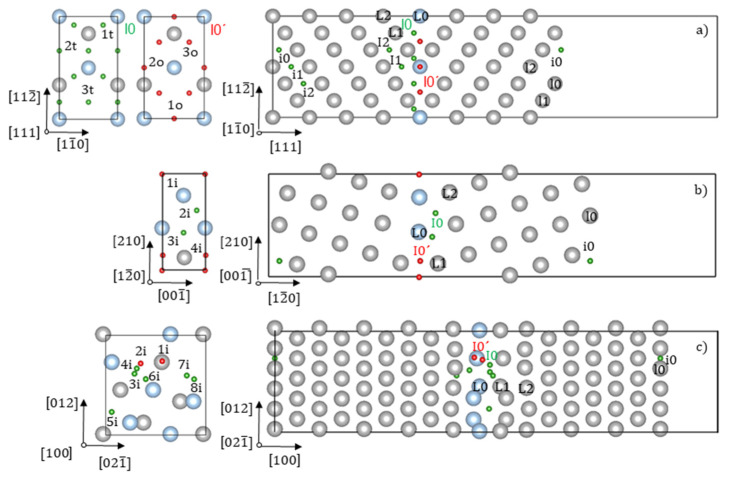
Schematic structures of (**a**) the coincident site lattice (CSL) Σ3(111) [1,2,3,4,5,6,7,8,9,10] grain boundary (GB) and (111) free surface (FS); (**b**) the CSL Σ5 (012) [100] GB and (012) FS; (**c**) the CSL Σ5 (100) [001] GB and (001) FS used in this work. Capital/not capital letters of the numbers of layers and H positions refer to GB/FS, respectively. The red spheres correspond to H interstitial sites located in the GB plane. The green spheres correspond to H interstitial sites located outside the GB plane. The view is normal to the GB planes; and no labels are used for the demonstration of the possible tetrahedral and octahedral sites of H in the I0 and I0´ layers for the case of Σ3(111) [1,2,3,4,5,6,7,8,9,10] and no labels for the H positions in the cases of Σ5 (012) [100] and Σ5 (100) [001]. I0 and I0´ labels are referred to the first layer of H located directly at the GB layer and the next to GB layer, which correspond to the octahedral and tetrahedral sites in the case of Σ3(111) [1,2,3,4,5,6,7,8,9,10]. The blue spheres correspond to the GB layers.

**Figure 3 materials-13-02288-f003:**
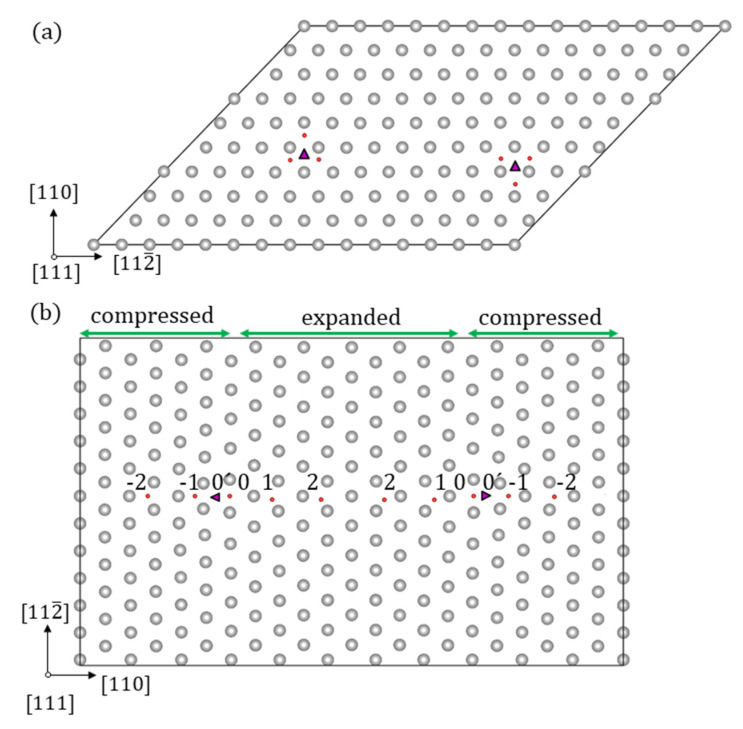
(**a**) ½<111> screw and (**b**) M111 mixed dislocations. The location of the dislocation core is marked as a purple triangle. Initial H atom positons are marked with the red spheres. The digits −2, −1, 0, 1, 2 are the numbers of H positions and correspond to Figure 7. 0´ H position is additionally considered one in the dislocation core, but it was found to be less energetically preferable during the atomic relaxation procedure and therefore is not shown in the H profile in Figure 7.

**Figure 4 materials-13-02288-f004:**
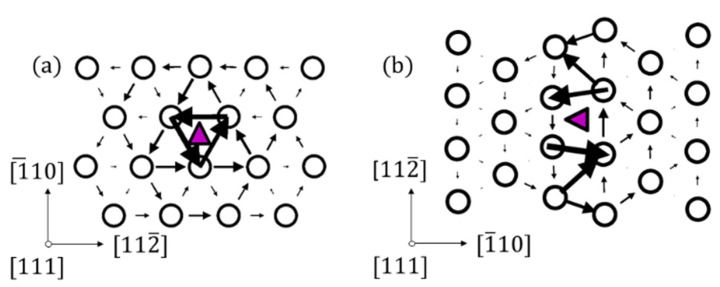
(**a**) ½<111> screw and (**b**) M111 mixed dislocations. The location of the dislocation core is marked as a purple triangle. The [111] (screw) component of the relative displacement of the neighboring atoms produced by the dislocation is depicted as an arrow between them.

**Figure 5 materials-13-02288-f005:**
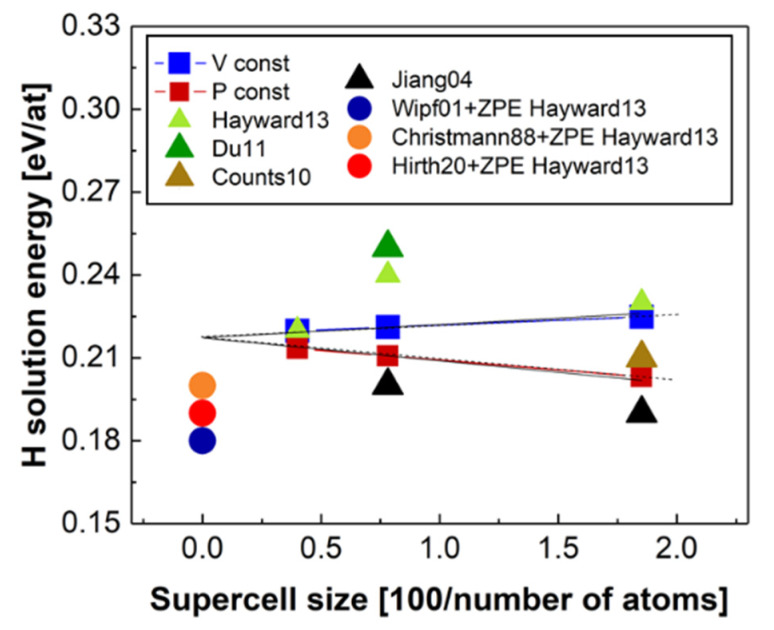
Dependence of the TS-H-interstitial formation energy on the supercell size at constant volume (blue line, blue squares) and at constant pressure (red line, red squares). The results of the present calculations are compared to other theoretical [62,78,107,113] data marked as triangles and experimental data extrapolated to 0 K (ZPE corrected) [110,111,112] and marked as circles.

**Figure 6 materials-13-02288-f006:**
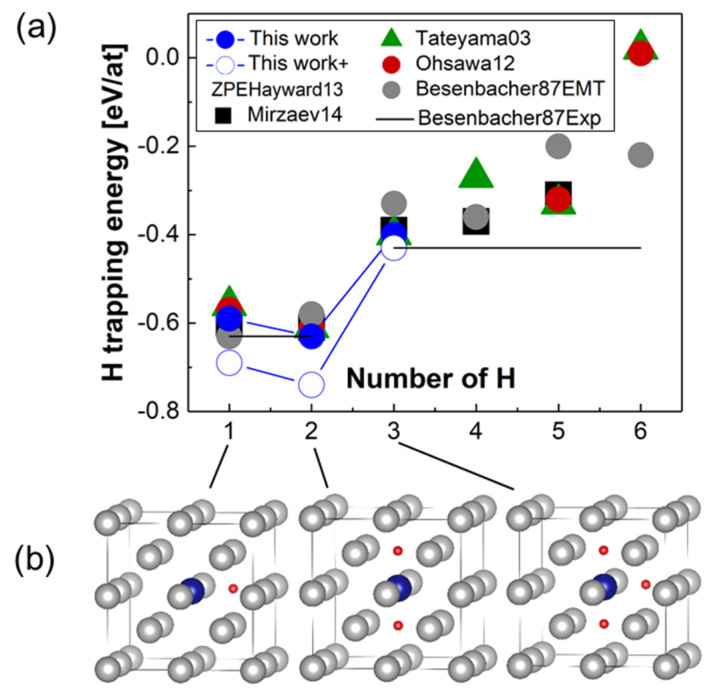
(**a**) Dependence of the H-trapping energy on the number of H atoms (n) in the nH-V cluster. The results of the present calculations are compared to other theoretical data [12,24,114] and experimental results [115,116]. (**b**) Structures of H-vacancy clusters are shown in the bottom panel. The vacancy is marked as a blue circle. Hydrogen atoms are shown with small red circles.

**Figure 7 materials-13-02288-f007:**
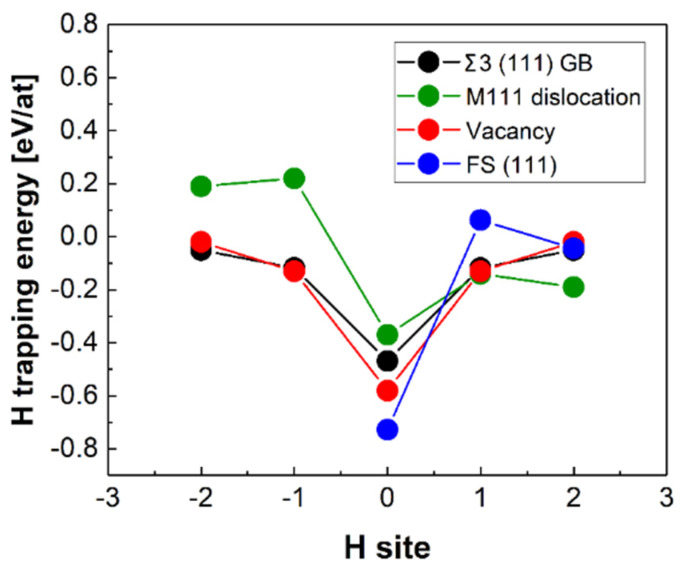
Hydrogen trapping profiles for Σ 3 (111) GB, (111) FS, mixed 111 dislocation and a vacancy. The trapping energies are presented relative to the geometrical centre of each defect indicated by 0. The considered trapping sites are located at the first, second and third atomic planes away from the corresponding defects, as indicated in Figure 2a and Figure 3b. In the case of a vacancy, the next nearest neighbour TS positions are shown. Minus signs refer either to mirrored or to compressed (in the case of M111 dislocation) crystallographic directions.

**Figure 8 materials-13-02288-f008:**
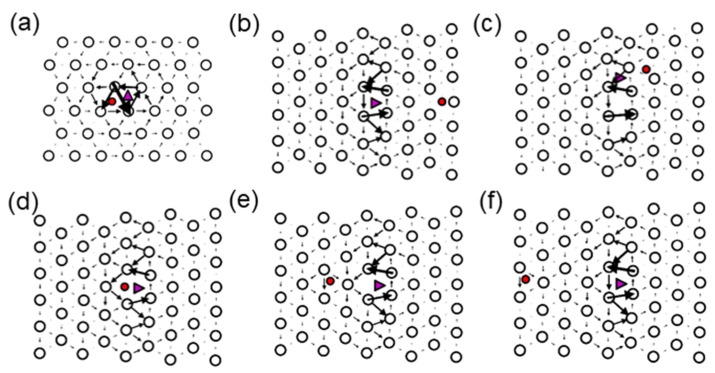
The locations of H sites (**a**) in the computational cell with two ½<111> screw dislocations, (**b**–**f**) corresponding to the sites from −2 to 2 from Figure 3b in the cell with two mixed 111 dislocations. The location of the dislocation core is marked as a purple triangle; the final positions of H after the optimisation are shown as the red circles. The [111] component of the relative displacement of the neighboring atoms produced by the dislocation is depicted as an arrow between them.

**Figure 9 materials-13-02288-f009:**
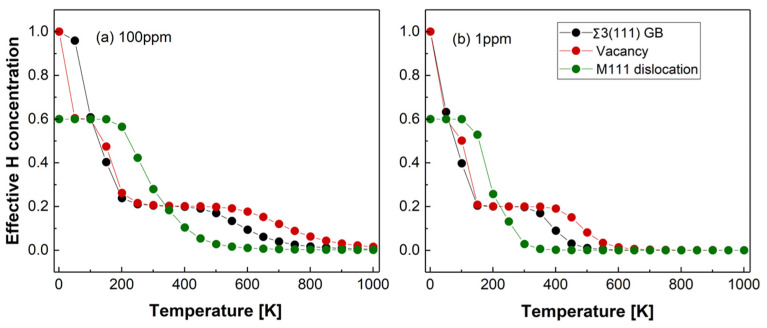
The temperature dependence of the H concentration at Σ3 (111) GB, M111 dislocation and a vacancy (**a**) for H bulk concentration 100 ppm, (**b**) for H bulk concentration 1 ppm.

**Table 1 materials-13-02288-t001:** Lattice parameter (a) and bulk modulus (B) and magnetic moment (μ) of bcc Fe.

Source	a, Å	B, GPa	µ, µ_B_
This work, PBE	2.831	181	2.19
Material project DFT [61]	2.847	182	2.33
Sanchez08 DFT, PBE [33]	2.815	175	2.25
Hayward13 DFT,PBE [62]	2.834	174	2.20
Rayne61 Exp. 3.2 K [69]	-	173	-
Söderlind00 DFT [71]	2.836	195	-
Haas09 DFT [63]	2.833	-	-
Haas09 Exp. [63]	2.853	-	-

**Table 2 materials-13-02288-t002:** Formation energies of a vacancy; ½<111> screw and (b) M111 mixed dislocations; (111), (100) and (012) FS; Σ3 (111) [1,2,3,4,5,6,7,8,9,10], Σ5 (100) [100] and Σ5 (012) [100] GBs and the corresponding W_sep_ of bcc Fe. The results are compared to the available theoretical [4,18,72,74,75,76,77,78,79,98,99,100,101,102,103] and experimental [64,80,94,95,96,97,104,105] data.

Characteristic	Defect Type	This Work	DFT Studies	Experimental
Formation energy, e	Vacancy	2.02	1.93, 1.95, 2.01 [72], 1.86, 2.06, 2.16 [75], 2.14 [76], 2.15 [73,74,77], 2.17 [77,78], 2.37 [18], 2.39 [79]	1.4 [94], 1.5 [95], 1.6 [96], 1.61–1.75 [97], 1.7 [64], 2.00 [80]
FS energy, J/m^2^	(111) FS	2.67	2.52 [106], 2.69 [4], 2.69 [98], 2.65 [74], 2.71, 3.23 [103]	2.42 [104], 2.48 [105]
	(012) FS	2.44		
	(100) FS	2.94	2.55, 3.06 [103], 2.29 [106]	
GB energy, J/m^2^	Σ3 (111) [1,2,3,4,5,6,7,8,9,10] GB	1.60	1.57 [102], 1.52 [4], 1.66 [74], 1.46 [100], 1.61 [99], 1.57 [101], 1.79 [103]	-
	Σ5 (012) [100]	1.60	2.00 [102], 1.64 [101], 1.83 [13]	-
	Σ5 (100) [001]	2.01	2.12 [101], 2.20 [103]	-
Work of separation, J/m^2^	Σ3 (111) [1,2,3,4,5,6,7,8,9,10]	3.76	3.86 [4], 3.65 [74], 3.78 [102], 4.60 [103]	-
	Σ5 (012) [100]	2.88	3.19 [102]	-
	Σ5 (100) [001]	3.86	3.90 [103]	-

**Table 3 materials-13-02288-t003:** Solution energies of H in pure Fe. The results are compared to the available theoretical [62,78,79,107,108] and experimental [110,111,112] data. The experimental data extrapolated to 0 K with ZPE [62] excluded are shown in parenthesis.

Type of H Site	Solution Energy, eV		
	This Work	Theoretical	Experimental
Interstitital tetrahedral	0.23 (4 × 4 × 4 cell) 0.22 (dilute limit)	0.19 [107], 0.21 [78], 0.23 [62], 0.27 [79]	0.30 (0.20) [110,111], 0.28 (0.18) [112]
Interstitital octahedral	0.37 (4 × 4 × 4 cell)	0.26 [62], 0.32 [107], 0.34 [78], 0.35 [79]	-
substitutional	2.54 (4 × 4 × 4 cell)	2.53 [108], 2.61 [78]	-

**Table 4 materials-13-02288-t004:** H-trapping energies in Fe (in eV/at). Literature data [12,13,14,15,16,17,18,23,27] and the results of this work. Zero point energy correction added to the 0 K results has been taken from references [17,27,62], and it is shown in the parentheses.

Type of Defect	Literature Data at 0 K	Literature Data at 0 K + ZPE (Defect +H)	Method	This Work	This Work + ZPE	Experimental Data
**Vacancy**			DFT, PBE			
H1V	–0.69 [117], −0.57 [114], −0.6 [12], −0.5 [62]	−0.56 [24], −0.62 [62]	DFT PW91 [12] DFT PBE [13,14,22,57,104]	−0.58	−0.70 (−0.12) [62]	−0.63 [115]
H2V	−0.61 [12,24,114], -0.54 [62]	−0.65 [62]		−0.63	−0.74 (−0.11) [62]	
H3V	−0.40 [24,114], −0.39 [12],−0.34 [62]	−0.38 [62]		−0.39	−0.43 (−0.04) [62]	−0.43 [115]
H4V	−0.27 [24], −0.36 [114], −0.37 [12], −0.30 [62]	−0.35 [62]				
H5V	−0.33 [24], −0.32 [114], −0.31 [12], −0.27 [62]	−0.27 [62]				
H6V	0.02 [24], 0.01 [114], 0.043 [62]	−0.045 [62]				
**GB**						
Tilt Σ3 (111)	−0.39 [107]	−0.58 [27]	DFT PBE	−0.47	−0.57 (−0.1) [27]	−0.18 [118] −0.28 [119] −0.61 [120]
Tilt Σ5 (012)	−0.81 [107]			−0.42		
Tilt Σ5 (013)	−0.43 [107]					
Tilt Σ9 (1/2 11)	−0.29 [15]		TB			
Tilt Σ13 (1/3 11)	−0.27 [15]					
Tilt Σ17 (1/4 11)	−0.32 [15]					
Twist Σ3(110)	−0.26 [17]					
Twist Σ5 (100)	-			−0.57		
Twist Σ9(110)	−0.68 [17]					
Twist Σ11(110)	−0.83 [17]					
Twist Σ17(110)	−0.95 [17]					
**Dislocation**						
Edge	−0.47[16]		QM/MM			−0.28 [118] −0.20 [121] −0.31 [122] −0.25 [123]
Screw1/2 <111>	−0.27 [19,20], 0.2 to −0.3 [124], −0.26 [23]	−0.32 [17]	QM/MM [19], DFT, PBE [20,125], MD [23]	−0.21	−0.26 (−0.05 [17])	
Mixed <111>	~−0.3 [124]	-	DFT PBE	−0.37		

**Table 5 materials-13-02288-t005:** Partial cohesive energy and strengthening energies, given in eV/at.

Characteristic	This Work	Literature
χ bulk	−3.34	-
η Σ3 (111) [1,2,3,4,5,6,7,8,9,10]	0.26	0.26–0.41 [10,13,27,125]
η Σ5 (012) [100]	0.41	0.07 [13]
η Σ5 (100) [001]	0.05	-

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
