# Peer review of "Hydrogen Trapping in bcc Iron"

_materials, 2020, doi:10.3390/ma13102288_

Round 1

Reviewer 1 Report

The manuscript titled "Hydrogen Trapping in bcc Iron " is well rewritten. The author has explained hydrogen trapping in bulk lattice, at vacancies, dislocations and grain boundaries by using DFT calculations. The theoretical basis is sufficient and the calculation is reasonable. Few things that can be modified to increase the more precise understanding of the manuscript.

1) Please, check and revise the grammar carefully. (e.g. line 59 and 62 “at. al.”)

2) Lines (68-73) and (314-315) text format problem.

3) From figure 9 we can see that, at the room temperature +/- 100K (approximately 200-400 K), hydrogen concentration at dislocations shows a much faster decreasing tendency with temperature increasing from 200 K to 400 K, while hydrogen concentration at vacancies and grain boundaries remains basically unchanged, how to explain the phenomenon and what does it mean for suppression of hydrogen embrittlement in steel.

Author Response

We would like to thank the referee for his useful feedback and positive review of our work. Please find our answers and clarifications below. All changes and corrections introduced in the new version of the manuscript are highlighted in blue.

1,2) “Please, check and revise the grammar carefully. (e.g. line 59 and 62 “at. al.”) Lines (68-73) and (314-315) text format problem”.

All required format changes were done.

3) From figure 9 we can see that, at the room temperature +/- 100K (approximately 200-400 K), hydrogen concentration at dislocations shows a much faster decreasing tendency with temperature increasing from 200 K to 400 K, while hydrogen concentration at vacancies and grain boundaries remains basically unchanged, how to explain the phenomenon and what does it mean for suppression of hydrogen embrittlement in steel”.

We agree with the reviewer that the figure 9 should be discussed more thoroughly. The following discussion has been added to the manuscript:

Changes to the manuscript:

 “At the room temperature +/- 100K (approximately 200-400 K), H concentration at dislocations shows a much faster decreasing tendency (at both 1 and 100ppm H in the bulk), while H concentration at vacancies and GBs remains basically unchanged. It is related to the different shape of the H trapping profile for the dislocation in comparison to H trapping profiles at the GB and vacancy shown in figure 7. This result indicates that H trapping at the M111 type of dislocations can be more sensitive to the bulk H concentration and temperature changes than trapping at GBs and vacancies, which is an important aspect of the HELP mechanism of HE.”

Please find the revised manuscript as the attachment.

Best regards, Anastasiia Kholtobina

Reviewer 2 Report

I attentively read the paper of  Anastasiia S. Kholtobina, Reinhard Pippan, Lorenz Romaner, Daniel Scheiber, Werner Ecker, and Vsevolod I. Razumovskiy “ Hydrogen Trapping in bcc Iron”. The authors have systematically investigated Hydrogen trapping in the bulk lattice and at all typical defects in bcc Fe by means of the same methodology of DFT calculations at 0 K.

The paper is thorough and well-written, and the main message, well emphasized, can be useful for to the community of researches in the field of Hydrogen embrittlement of steels. I can conclude that this is an accurate study performed at high level and the paper is recommended for publication in Materials.

I only have a minor concern.

  • 45 “ which are also higher than those of” replace with “ which are also lower than those of”
  • 108  “separated by 15, 7 and 7 Å of vacuum as schematically shown in Figure 2” Why didn't the authors use 15 A for all superlattices? 7 A may not be sufficient to prevent interaction between films through the vacuum layer.
  • The value of the digits -2, -1, 0, 1, 2 in figure 3b is not obvious.
  • In table 1, the meaning of the @ symbol in the line " Rayne61 Exp. @ 3.2 K [69]”
  • According to the data in table 2, the calculated energy value 2.44 for the surface (012) is significantly lower than the value 2.67 for the surface (111). How do the authors explain this result? The density of atoms on the surface (111) is lower than on the surface (012)? Why is the calculated energy value for the surface (110) not given? The surface (110) has the highest density of atoms in the BCC lattice.
  • In paragraph 5.2, the authors present the results of calculations "The effective H concentration at the vacancy, M111 dislocation and Σ3 (111) GB". At a given average concentration, the distribution of hydrogen across defects depends not only on the binding energy of hydrogen with these defects, but also on the concentration of defects. In this case, it depends on the vacancy concentration, dislocation density, and grain boundary area per unit volume. How did the authors evaluate the concentration of defects?
  • 341 “Hydrogen site numbering along to the x axis corresponds to Figures 6 (dislocation) and 7 (GB).” References to drawings illustrating the positions of hydrogen atoms are not clear.

Author Response

We would like to thank the referee for his useful feedback and positive review of our work. Please find our answers and clarifications below. All changes and corrections introduced in the new version of the manuscript are highlighted in blue.

  • “separated by 15, 7 and 7 Å of vacuum as schematically shown in Figure 2” Why didn't the authors use 15 A for all superlattices? 7 A may not be sufficient to prevent interaction between films through the vacuum layer.

Changes to the manuscript:

“Special CSL model GBs Σ3(111)[1-10], Σ5 (012)[100] and Σ5 (100)[001] were modelled by supercells containing 49, 30, 44 atomic layers of Fe (2, 1 and 5 atoms per layer) separated by 15, 7 and 7 Å of vacuum, which were tested to be sufficient within 0.01 eV/at error at most, as schematically shown in Figure 2“.

  • „The value of the digits -2, -1, 0, 1, 2 in figure 3b is not obvious“.

Changes to the capture of figure 3 :

“The digits -2, -1, 0, 1, 2 are the numbers of H positions and correspond to Figure 7. 0´ H position is additionally considered one in the dislocation core, but it was found to be less energetically preferable during the atomic relaxation procedure and therefore is not shown in the H profile in Figure 7”.

  • “According to the data in table 2, the calculated energy value 2.44 for the surface (012) is significantly lower than the value 2.67 for the surface (111). How do the authors explain this result? The density of atoms on the surface (111) is lower than on the surface (012)? Why is the calculated energy value for the surface (110) not given? The surface (110) has the highest density of atoms in the BCC lattice”.

 The planar density of atoms within (111) surface in BCC  is higher than that of the (012) surface and therefore the energy of the (111) plane is higher. This result is also in agreement with an earlier study by Vitos et al. [1], where it was shown that (111) surface is the highest-energy surface for bcc metals. The surface energies in our paper are given only for the free surfaces, which are formed after considered GB separation into free surfaces. Since we have used (111) GB, (100) GB and (012) GB the surface energy for (110) FS was not calculated in the framework of this study.

  • “In paragraph 5.2, the authors present the results of calculations "The effective H concentration at the vacancy, M111 dislocation and Σ3 (111) GB". At a given average concentration, the distribution of hydrogen across defects depends not only on the binding energy of hydrogen with these defects, but also on the concentration of defects. In this case, it depends on the vacancy concentration, dislocation density, and grain boundary area per unit volume. How did the authors evaluate the concentration of defects?”

The referee is right: this result will depend on trap densities as well as the ratio between the available H and available trapping sites may lead to abundance/shortage of the segregation element (H) in the system. This aspect of the problem could be included within the framework of a phenomenological model published by some of the co-authors of this paper recently [2]. However, this point goes beyond the scope of the present study that uses a classical representation of the McLean segregation isotherm where the effect of trap density dependency changes is not taken into account[3].

Please find the revised manuscript as the attachment.

Best regards, Anastasiia Kholtobina

References

  1. Vitos, L.; Ruban, A. V; Skriver, H.L.; Kolla, J. The surface energy of metals. Surf. Sci. 1998, 411, 186–202.
  2. Drexler, A.; Depover, T.; Leitner, S.; Verbeken, K.; Ecker, W. Microstructural based hydrogen diffusion and trapping models applied to Fe–C-X alloys. J. Alloys Compd. 2020, 154057, doi:10.1016/j.jallcom.2020.154057.
  3. McLean, D. Grain boundaries in metals; Clarendon Press: Oxford, 1957.

Reviewer 3 Report

This manuscript investigated the hydrogen trapping in iron using density functional theory calculations. Most of calculations were conducted at 0 K, and were compared with previous reports and experiments. Authors should clearly define the difference between this DFT calculations and previous calculations and experiments, and advantageous of this method in introduction section. The comments are as follows.

  1. In p. 2, line 57, “MD” should be “Molecular Dynamics (MD)”. When authors use abbreviations first time, please define the abbreviations.
  2. In p. 2, line 59, “Lu at. al.” should be “Lu et al.”.
  3. In p. 2, line 86, what is “PBE”?
  4. In p. 5, lines 166 and 167, “E are” should be “E is”.
  5. In Table 1, please define “a”, “B” and “m”.
  6. Tables 2 and 5 were separated in two pages. These were difficult to check the Tables. Please indicate each Table in one page.
  7. In lines 212, 222, 231, 259, 259, 276, 308, 342 and 385, these are titles. Authors should number them, such as (1) Ground state properties of the bulk Fe or 4.1.1.1 Ground state properties of the bulk Fe.
  8. In p. 9, line 283, what is “SC”?
  9. In p. 10, line 314, authors used the unit both “eV/atom” and “eV/at.”. Please use the unit either “eV/atom” or “eV/at.”
  10. About Figure 7 (and other figures), authors should explain the figure more politely. It is difficult to understand the results for readers.

Author Response

We would like to thank the referee for his useful feedback and positive review of our work. Please find our answers and clarifications below. All changes and corrections introduced in the new version of the manuscript are highlighted in blue.

All required format changes have been implemented. We would like to thank the referee for the constructive comments.

  • Authors should clearly define the difference between this DFT calculations and previous calculations and experiments, and advantageous of this method in introduction section”.

Changes to the manuscript:

„In this work, we perform a systematic DFT investigation of H trapping in the bulk lattice, at vacancies, dislocations and special GB in ferromagnetic bcc Fe. “In comparison to previous theoretical studies using different variations of the generalized gradient approximation (GGA) for the exchange-correlation functional or even less precise tight-binding (TB) approximation calculations, this study provides a consistent set of  H trapping energies  obtained within the same methodological approach that provides a set of energies for qualitative and quantitative interpretation of experimental (for instance thermal desorption spectroscopy) data.“

  • About Figure 7 (and other figures), authors should explain the figure more politely. It is difficult to understand the results for readers.

Changes to the capture of figure 7 :

“The trapping energies are presented relative to the geometrical centre of each defect indicated by 0. The considered trapping sites located at 1st, 2nd and 3rd atomic planes away from the corresponding defects, as indicated in Figures 2a and 3b. In the case of a vacancy, the next nearest neighbour TS positions are shown. Minus signs refer either to mirrored or to compressed (in the case of M111 dislocation) crystallographic directions. ”

Changes to the capture of figure 3 :

“The digits -2, -1, 0, 1, 2 are the numbers of H positions and correspond to Figure 7. 0´ H position is additionally considered one in the dislocation core, but it was found to be less enrgetically preferable during the atomic relaxation procedure and therefore is not shown in the H profile in Figure 7.”

Please find the revised manuscript as the attachment.

Best regards, Anastasiia Kholtobina
